# An Expedient Catalytic Process to Obtain Solketal from Biobased Glycerol

**Fabrizio Roncaglia** [1,*] **, Luca Forti** [2] **, Sara D'Anna** [1] **and Laura Maletti** [1]

1   Department of Chemical and Geological Sciences, University of Modena and Reggio Emilia,
    via G. Campi 103, 41125 Modena, Italy; sara.danna1993@outlook.it (S.D.); laura.maletti@unimore.it (L.M.)
2   Department of Life Sciences, University of Modena and Reggio Emilia, via G. Campi 103, 41125 Modena,
    Italy; luca.forti@unimore.it
*   Correspondence: fabrizio.roncaglia@unimore.it

**Abstract:** Developing simple and effective chemistry able to convert industrial waste streams into valuable chemicals is a primary contributor to sustainable development. Working in the context of biodiesel production, we found that plain bisulfate on silica (SSANa, 3.0 mmol/g) proved to be an optimal catalyst to convert glycerol into solketal. With the assistance of a proper anhydrification technique, isolated yields of 96% were achieved working in mild conditions, on 100 g scale.

**Keywords:** biodiesel; glycerol; silica sulfuric acid; acetalization; solketal

## 1. Introduction

The progressive depletion of fossil carbon resources and environmental concerns related to their exploitation raise serious doubts about the sustainability of our current development model. This encouraged extensive research to address a reduction of our dependence from hydrocarbon-based resources. But humanity's hunger for energy has reached unprecedented levels and our current need for carbon-based materials and fuels will continue to grow in the next decades. To support our socio-economic development a great deal of attention was recently devoted to the study and implementation of biorefineries. Biorefineries are production plants where renewable biomass is economically converted into carbon-based products, including materials, fuels, and energy [1]. Carbon dioxide fixation during plant growth can considerably reduce the impact on carbon balance, when a vegetable source is exploited. Additional drive to biorefineries expansion is the price volatility induced by the awareness of the limited nature of fossil resources.

Biodiesel biorefineries focus on the production of hydrocarbon mixtures that can be blended with regular diesel fuel, starting from vegetable or animal fats. Natural fats (triglycerides) are subjected to acid or base catalyzed transesterification in the presence of an excess of a short chain alcohol (Figure 1), to give a mixture of fatty acid methyl (or ethyl) esters, that is biodiesel. Glycerol (**gly**) is also co-produced and is usually considered a waste stream, especially in small-scale productions where further refining costs are not justified. The present crude **gly** price is as low as 0.04–0.09 $/lb and it is expected to decrease further, according to the growing industrial availability; predicted global production is 41.9 billion liters by 2020 [2]. Glycerol can have some direct uses in the pharma, food, cosmetic, or polymer industries, but the presence of three vicinal hydroxyl groups in its structure induces high boiling point and viscosity, low miscibility with organics, as well as chemoselectivity issues when used as building block (primary vs. secondary hydroxyl) [3].

As a result, the identification of simple and effective processes able to convert **gly** into value-added chemicals unveil tremendous potential and can give substantial contribution to the sustainability of the whole biodiesel industry [2,4]. One of the most promising routes involves the production of cyclic acetals from the condensation of **gly** and carbonyl

compounds, with applications as fuel oxygenates or chemical intermediates [2,4,5]. Glycerol acetals can work as octane enhancer into regular gasoline, or can reduce particulate emissions of diesel fuel [6–8]. Other notable uses are in the field of polymers, where the said selectivity issue is solved by the acetal protecting group, giving the opportunity of a precise inclusion as a monomer [3,9].

**Figure 1.** Biodiesel production through transesterification.

Solketal (**1**, Figure 2), the acetal obtained from glycerol and acetone, is well described in the literature and commonly prepared in homogeneous or heterogeneous processes catalyzed by mineral acids (e.g., $H_2SO_4$), organic acids (p-toluenesulfonic acid), Lewis acids (e.g., $SnCl_2$), synthetic acidic resins (e.g., Amberlyst 15®), and/or zeolites (e.g., beta-zeolite) [10]. As acetone is involved, the process shows high selectivity toward the 5-membered 1,3-dioxolan ring (**1**), while the isomeric 1,3-dioxane **1a** is formed only at trace levels.

**Figure 2.** Solketal preparation through acetalization of **gly**.

Irrespective from the employed catalyst, this preference mainly depends on the substitution degree of the carbonyl compound. Reaction of **gly** with an aldehyde regularly results in high selectivities for 1,3-dioxa acetals [11], while the two carbonyl substituents of ketones induce unfavorable 1,3-diaxial repulsions inside the six-membered ring [3]. Only when the two substituents display enough steric distinction, like in 4-methyl-2-pentanone for instance, significant amounts of six-membered ring acetal can be obtained from ketones [12,13].

In view of our interest in designing novel methods able to convert industrial waste streams into valuable and sustainable chemicals, we started extensive bibliographic evaluation in order to develop optimal conditions to convert **gly** into **1**. Both the increase of acetone amount and the removal of water are expected to favorably shift the equilibrium of Figure 2 toward **1**. Of the two, the first option involves simpler operation because of the easy separation of low boiling acetone at the end of the process while water removal introduces more sophistication, including the need for desiccants and/or azeotropic distillation. An intriguing issue suggested in the recent literature is the possibility to avoid water removal through the choice of a proper catalyst, able to tolerate small amounts of water without deactivation [10]. In order to ascertain this opportunity, we started an experimentation around $SnCl_2 \cdot 2H_2O$ catalyst, projected to replicate at best the litera-

ture conditions [12,13]. Unfortunately, notwithstanding several trials, what we get was a disappointing 25% isolated yield (entry 1, Table 1).

**Table 1.** Preliminary catalyst evaluation.

| Entry | Catalyst | Acetone Anhyd. [a] | Added Desiccant | Isolated Yield of 1 (%) |
|---|---|---|---|---|
| 1 | $SnCl_2 \cdot 2H_2O$ | n | - | 25 |
| 2 | $SnCl_2 \cdot 2H_2O$ | y | - | 41 |
| 3 | $SnCl_2 \cdot 2H_2O$ | y | MS (after 30′) | 70 |
| 4 | - | y | MS | 0 |
| 5 | "dry" SSA (2.63 mmol/g) | y | MS (after 30′) | 75 |
| 6 | "wet" SSA (2.63 mmol/g) | y | MS (after 30′) | 75 |
| 7 | $SiO_2$ | y | MS (after 30′) | 0 |

Reaction conditions: **gly** (2 g), acetone (6.4 mL, 4 eq), $SnCl_2 \cdot 2H_2O$ (74 mg, 1.5 mol%), MS (powder, 4 Å, 120 mg), 4 h 30 min, 23 °C; [a] n = no anhydrification; y = acetone pre-anhydrified with $CaCl_2$ for 24 h.

This prompted us to reconsider some experimental conditions, including water removal and, as expected, the simple anhydrification of acetone (on anhydrous $CaCl_2$ for 24 h) proved very effective to improve the result (entry 2). In addition, further inclusion of 4 Å molecular sieves (MS) brought the yield to a respectable 70% (entry 3). This did not depend to any catalytic activity of MS, as made clear in entry n. 4, where tin chloride was omitted. Curiously enough, it was necessary to delay the introduction of MS thirty minutes after the reaction start as the concurrent insertion of **gly** and MS resulted in very low yields and produced a viscous residue, a possible consequence of a catalytic oligomerization of glycerol [14]. Anyway, these data suggested that some anhydrification is probably essential, even with a supposedly water-tolerant catalyst like tin chloride. In consideration that water is a typical contaminant of raw glycerol coming from biodiesel industry [15,16], the use of a raw substrate can only make things worse.

The failure of $SnCl_2$ to support the process in absence of anhydrification brought our attention toward its significant toxicity (H332, H314, H317, H335, H341, H361, H373, H410), that couple with a high solubility in the reaction media, giving rise to troublesome separation from the products. In the search for a less problematic catalyst, we were attracted by the use of a simple Brønsted acid, as most of them demonstrated both a good catalytic activity in acetalizations and very low toxicity. Unluckily, high solubility (difficult separation/recovery) and plant corrosion issues hampered their clear use [17–19]. These considerations brought us to the conclusion that having a heterogeneous catalyst is the most desirable choice.

Beyond the popular acidic clays, zeolites or synthetic resins, an appealing option appeared to be the use of acidic species over an inert organic [18], or inorganic support [16]. Despite the basic idea is not particularly innovative, it is the specific implementation that can confer its attractiveness. In fact, looking at the recently proposed methods, most of them suffer from over-sophistication due to the prevalence of a "basic research—driven" approach, with marginalization of the applied side [20,21]. This can be detected by the presence of laborious and energy/time-consuming operations possibly including the catalyst preparation or regeneration, making questionable the overall applicability of the method. This being a high-volume low-value sector, the use of a "use-inspired basic research" approach is particularly appropriate. The term, related to the simultaneous consideration of both the understanding/innovation and the simplicity/feasibility sides, refers to Louis Pasteur's studies, in particular to its Quadrant model [20,21].

Therefore, dealing with simple/elegant solutions, suitable to industrial consideration, we were surprised to note that simple acids supported on silica received little attention in this field. Only a few reports considered the use of sulfuric acid on silica for glycerol acetalization [22–25], but the current technique require high temperature (130 °C), microwaves, or a large excess of acetone (20 eq) to gain acceptable conversions, so further study was needed to reveal its potential.

## 2. Materials and Methods

### 2.1. General Information

Solvents and reagents were commercial grade and used as is, except acetone (water $\leq$ 0.2% *w/w*, from Carlo Erba) that was dried on $CaCl_2$ (anhydrous, purchased from Merck, 50 g/L for 24 h) and kept well closed. The employed silica was a standard chromatographic grade Silica Gel 60, having 63–200 μm particle size (70–230 mesh, from Merck). Molecular sieves powder (2.5 μm, 4 Å porosity, activated) was purchased from Merck, while molecular sieves beads (8–12 mesh, 4 Å porosity, grade 514, W.R. Grace & Co., Columbia, MD, USA) were from Fisher Scientific. IR spectra were recorded on a Jasco spectrometer (FT/IR 4700) and processed with Spectragryph 1.2 software; samples put in KBr pellets. Protonic NMR spectra were acquired with a Bruker Avance 400 spectrometer.

### 2.2. General Procedure for "Wet" Preparation of Supported Catalysts

In a 25 mL round bottom flask containing a magnetic stirring bar, the catalyst (7.5 mL of an aqueous solution containing x mol/L) and $SiO_2$ (3.75 g) were inserted, then the mixture was stirred for 30 min. Once evaporated to dryness at the rotavapor, the solid residue (free-flowing powder) was oven-dried at 130 °C overnight, giving the supported catalyst on silica. To obtain a certain loading (y [mmol/g]), the molarity of the employed solution (x) can be calculated with the following expression:

$$x(mol/L) = \frac{500y}{100 - y \cdot mw}$$

where mw [g/mol] is the molecular mass of the catalyst under consideration

### 2.3. Preparation of SSANa (3.0 mmol/g) Catalyst

Once the loading (y = 3.0 mmol/g) and the catalyst ($NaHSO_4$, with mw = 120.05 g/mol) are established, the concentration of the aqueous solution was calculated (x = 2.34 mol/L) and the catalyst was prepared following the procedure described in the previous section (2.2).

Hence, in a 25 mL round bottom flask containing a magnetic stirring bar, $NaHSO_4$ (2.34 M aqueous solution, 7.5 mL, 17.55 mmol) and $SiO_2$ (3.75 g) were inserted and the mixture was stirred for 30 min. Once evaporated to dryness at the rotavapor, the solid residue (free-flowing powder) was oven-dried at 130 °C overnight, giving SSANa (3.0 mmol/g) as a white powder (5.86 g).

### 2.4. Solketal Preparation (Small Scale)

To a stirred mixture of glycerol (2.00 g, 21.7 mmol) and acetone (dried on $CaCl_2$, 6.40 mL, 4 eq), inside a 10 mL vial with screw cap, the supported catalyst (1.5 mol%) was added at rt. After 30 min, MS (4 Å, powder, 120 mg) were added and the mixture was left under stirring for 4 h. The solid catalyst was filtered on a Gooch funnel (porosity 3) and the liquid phase was dried at the rotavapor. EtOAc (4 mL) and *n*-hexane (4 mL) were added to the residue, and the mixture was centrifuged (4000 rpm) for 5 min. A small amount of a lower phase, containing unreacted gly and water was detected. Removal of the volatiles from the organic phase left solketal (1) as a colorless liquid (2.15 g, 75%).

### 2.5. Solketal Preparation (Bigger Scale)

Glycerol (100 g, 1.086 mol) and acetone (dried on $CaCl_2$, 320 mL, 4 eq) were vigorously stirred inside a 1 L round bottom flask at rt, then SSANa (3.0 mmol/g, 5.50 g, 1.5 mol%) was inserted and the mixture was stirred for 25 min (the mixture changed from hazy to clear in few minutes). $CH_2Cl_2$ (600 mL) was inserted and a Dean-Stark trap, featuring a long arm funnel inside and filled with MS (4 Å, 3 mm beads, 75 g), was set up, together with an appropriate water-cooled condenser (see Figure S1). The flask was immersed in a pre-heated oil bath (around 55 °C) and refluxed for 3 h 30 min. The mixture was filtered to remove the solid catalyst, then the volatiles ($CH_2Cl_2$ and acetone) were recovered at the

rotavapor, leaving raw solketal (**1**) inside the reaction flask (colorless liquid, 137.8 g, 96%). Its high purity (>99% GC) makes it adequate for most applications ($^1$H-NMR spectra is reported in Figure S2). If desired, a purer version ($^1$H-NMR spectra showed in Figure S3) is easily attainable by distillation at reduced pressure (b.p. 77°C at 10 mm Hg).

## 3. Results and Discussion

Silica sulfuric acid ($H_2SO_4 \cdot SiO_2$ or SSA, also known as sulfuric acid adsorbed on silica) is a well-known solid acid catalyst with various applications in organic chemistry [26]. It can be obtained by mixing aqueous sulfuric acid with regular chromatography silica gel, followed by water removal at 130 °C (hereafter referred as "wet" SSA or simply SSA) [27–29], or by dropping chlorosulfonic acid directly on dry silica (referred as "dry" SSA") [30–32]. Its features like low cost, availability, low toxicity, and easy separation from the reaction media spurred us to study the application of the two popular SSA variants (typical loading 2.63 mmol -$SO_3H$/g) in the said acetalization process (Figure 2).

Working in the earlier defined experimental conditions, and excluded the potential contribution from the support alone (entry 7, Table 1), we detected an improved performance of each SSAs in comparison to $SnCl_2$ (entry 5 or 6 vs. entry 3, Table 1). This represents the first comparison in literature of the two SSA variants, showing that "wet" SSA can perform like the "dry" SSA, without showing any negative, moisture-related effects. The more sustainable "wet" SSA was then selected as our reference catalyst.

In continuation of our evaluation, a set of SSA featuring different $H_2SO_4$ loading on $SiO_2$ was prepared and tested, keeping constant the moles of acid with respect to gly (1.5 mol%). Within the investigated range, performances resulted almost independent from this parameter (Figure 3): the little preference for lower loading was rationalized in terms of a small contribution of the dry $SiO_2$ on general anhydrification. The loading value of 3.0 mmol of acid per g of silica was selected as optimal, corresponding to the use of 55 g of solid catalyst in the conversion of 1 kg of **gly**.

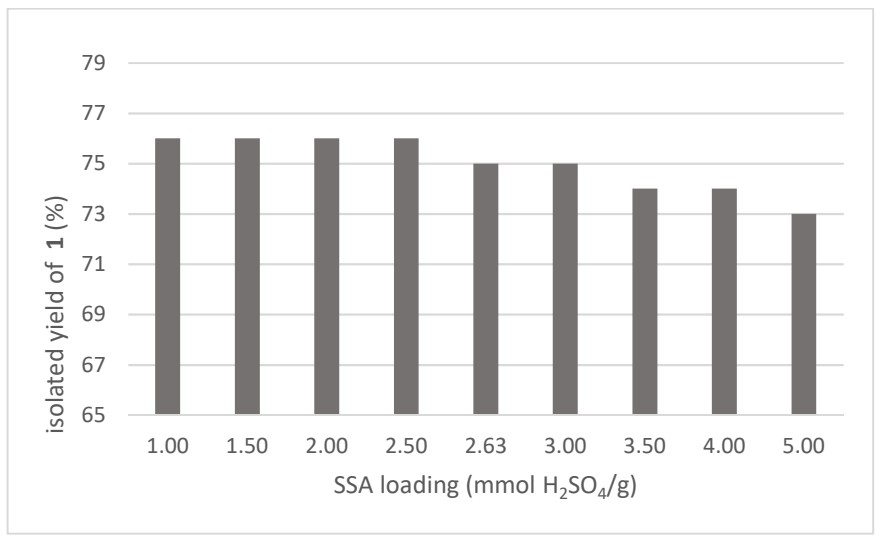

**Figure 3.** Performances of sulfuric acid adsorbed (SSA) with different $H_2SO_4$ loading (1.5 mol% respect to **gly**).

In the attempt to compare alternative Brønsted species, orthophosphoric acid was first considered but, when supported on silica ($H_3PO_4 \cdot SiO_2$, 3.0 mmol/g) it gave a disappointing result on acetalization of glycerol (27% isolated yield), suggesting the importance of the acid strength [4]. At this point, finding other candidates seemed not straightforward as monoprotic donors were expected to be devoid of catalytic activity, accordingly to the popular SSA representation based on sulfuric acid esters with surface silanols [22–25,27–32].

Anyway, going on with further bibliographic research, we were surprised to note the reported activity of silica perchloric acid (HClO$_4$·SiO$_2$) [33–36], or silica triflic (trifluoromethanesulfonic) acid [37–39], even in the acetalization processes. Potassium bisulfate on silica (KHSO$_4$·SiO$_2$) showed good activity too [40], although it was not tested in acetalization processes, until now. This information spurred us to prepare these last catalysts, consistently with published procedures, and test them within our reactive system. The data, collected in Table 2, confirmed a good activity for most of them. Despite the higher strength of the free acid, silica perchloric acid (SPCA, entry 1 Table 2) or silica triflic acid (STA, entry 2) did not improve the isolated yields in respect to SSA. Thus, according to a use-inspired basic research, we judged the employment of these "special" acids to be not justified. Perchloric acid, in particular, features known safety problems on concentration [36], so loading on silica was limited to 0.75 mmol/g [35].

**Table 2.** Evaluation of different silica supported acids.

| Entry | Catalyst | Isolated Yield of 1 (%) |
|---|---|---|
| 1 | SPCA (0.75 mmol/g) | 76 |
| 2 | STA (1.5 mmol/g) | 76 |
| 3 | SSANa (3.0 mmol/g) | 76 |
| 4 | SSAK (3.0 mmol/g) | 76 |
| 5 | SSAN (3.0 mmol/g) | 75 |
| 6 | SSAI (3.0 mmol/g) | 27 |

Reaction conditions: **gly** (2 g), acetone (dried on CaCl$_2$, 6.4 mL), catalyst (1.5 mol%), MS (powder, 4 Å, 120 mg) inserted after 30 min, 4 h 30 min, 23 °C. SPCA = silica perchloric acid; STA = silica triflic acid; SSANa silica bisulfate (sodium salt); SSAK silica bisulfate (potassium salt); SSAN silica bisulfate (ammonium salt); SSAI silica bisulfate (*N*-Me-imidazolium salt).

A series of bisulfates never employed in acetalization processes were also considered, namely: sodium (entry 3, Table 2), potassium (entry 4), ammonium (entry 5), and *N*-Me imidazolium (entry 6). Despite salification did not promise advantages on the acid strength, the observed performances were not inferior to that obtained with SSA (excluding the last one). Sodium or potassium bisulphates, in particular, displayed improved anchoring on silica surface, presumably due to the increased polarity. This was inferred by the absence of any liquid residue inside the container used for the preparation of SSANa (3.0 mmol/g) or SSAK (3.0 mmol/g), while tiny droplets of acidic liquid were often detected inside the walls of the glass container used in SSA (3.0 mmol/g) preparation, after the final drying step (overnight, 130 °C). The intrinsic lower acidity of these supported bisulfates can give potential advantage over SSA, as suggest less plant corrosion issues, in the event of partial leaching. The above considerations brought us to choose the less expensive one as the most convenient catalyst, i.e., SSANa (3.0 mmol/g).

In a final re-evaluation of the catalyst amount, we established that 1.5 mol% with respect to **gly** was an optimal value (see entries 1, 2, 3 of Table 3), and again, that the system highly benefits from the implementation of adequate anhydrification (entries 4–8 vs. entry 3). In absence of added desiccants (entry 4 of Table 3 vs. entry 2 of Table 1) the better water tolerance of SnCl$_2$·2H$_2$O with respect to SSANa (3.0 mmol/g) is detectable. But when adequate water removal is implemented, the superiority of the second is somewhat evident, both in performance (entry 2 of Table 3 vs. entry 3 of Table 1) and in easiness of separation from the reaction products; as well as in intrinsic toxicity.

**Table 3.** Optimization of SSANa (3.0 mmol/g) amount.

| Entry | Catalyst Amount (mol%) [a] | Desiccant (mg) | Isolated Yield of 1 (%) |
|:---:|:---:|:---:|:---:|
| 1 | 1.25 | MS powder (120) | 70 |
| 2 | 1.50 | MS powder (120) | 75 |
| 3 | 1.75 | MS powder (120) | 76 |
| 4 | 1.50 | - | 15 |
| 5 | 1.50 | $CaCl_2$ (150) | 31 |
| 6 | 1.50 | $MgSO_4$ (150) | 50 |
| 7 | 1.50 | $MgSO_4$ (300) | 51 |
| 8 | 1.50 | $Na_2SO_4$ (150) | 30 |

Reaction conditions: **gly** (2 g), acetone (dried on $CaCl_2$, 6.4 mL), SSANa (3.0 mmol/g), anhydrificant added after 30 min, 4 h, 23 °C. [a] mol% in respect to **gly**.

The study of the catalyst was completed by considering supported Lewis acids. Various articles suggest potential synergism between both Lewis and Brønsted acidity, the last coming from silica alone [41,42], or even by the deliberate addition of a second active species [43]. In the case of **gly** acetalization, we observed that when a Lewis acid was supported on silica by means of the wet method, its catalytic activity was greatly reduced (entry 1 Table 4 vs. entry 3 Table 1). The sequential deposition of a Brønsted acid followed by a Lewis acid [43], showed no particular advantage too (entries 4–7 Table 4) with respect to the plain Brønsted acid [44].

**Table 4.** Catalytic activity of supported Lewis acids.

| Entry | supported Catalyst (mmol/g) | Isolated Yield of 1 (%) |
|:---:|:---:|:---:|
| 1 | $SiO_2 \cdot SnCl_2$ (2.5) | 32 |
| 2 | $SiO_2 \cdot FeCl_3$ (2.5) | 21 |
| 3 | $SiO_2 \cdot ZnCl_2$ (2.5) | 0 |
| 4 | $SiO_2 \cdot H_3PO_4$ (3.0)$\cdot SnCl_2$ (1.5) | 24 |
| 5 | $SiO_2 \cdot H_3PO_4$ (3.0)$\cdot FeCl_3$ (1.5) | 18 |
| 6 | $SiO_2 \cdot H_2SO_4$ (3.0)$\cdot SnCl_2$ (1.5) | 74 |
| 7 | $SiO_2 \cdot H_2SO_4$ (3.0)$\cdot FeCl_3$ (1.5) | 72 |

Reaction conditions: **gly** (2 g), acetone (dried on $CaCl_2$, 6.4 mL), catalyst (1.5 mol%), MS (powder, 4 Å, 120 mg) added after 30 min, 4 h, 23 °C.

The infrared spectra of SSANa (3.0 mmol/g), in comparison to that of silica (oven dried at 130 °C, overnight), showed remarkable differences (Figure 4). The broad band at 3400 cm$^{-1}$ (relative to the O-H stretching of water and SO$_3$O-H group) together with the shoulder in the range 2900–2400 cm$^{-1}$ suggest strongly coordinated water via hydrogen bonds to silanols and -SO$_3$H moieties, indication confirmed by the increased H$_2$O bending band (~1635 cm$^{-1}$). The apparent splitting of the intense band from 1250 to 950 cm$^{-1}$ (assigned to asymmetric stretching vibrations of Si-O bonds) results from the superimposition of sulfate-stretching modes (asymmetric O=S=O (1252 cm$^{-1}$), symmetric O=S=O (1080 cm$^{-1}$), S=O (1024 cm$^{-1}$), and S-O (709 cm$^{-1}$)) that feature similar molecular weight of silicon tetrahedra [27,45]. Significative changes of the original silica vibrational structure are particularly evident from 900 cm$^{-1}$ to 500 cm$^{-1}$, while the 610 cm$^{-1}$ band could be assigned to $\nu_{ab}$ of HSO$_4$ group.

To the best of our knowledge, SSA structure has not been fully characterized, possibly due to its amorphous nature. Most of the literature reports depicts it as a sulfuric acid ester of surface silanols [22–25,27–32], despite this model is unable to explain the known activity of supported monoprotic acids, nor of bisulfates. Sulfuric esters may correspond to possible transient states during the formation of SSA (or SSANa), but then hardly represents the final form, because of the known hydrolytic instability of silyl sulfones [46–48]. A better even if approximate representation of the catalyst structure, proposed in Figure 5, involves a strongly adsorbed species that, together with some adsorbed water molecules, have the ability to exchange protons with the environment. In this model the peculiar role

of residual water is showed, probably responsible for the conserved activity of natively monovalent species. It also could have a prominent role in the apparent acidity levelling observed between bisulfate and sulfuric acid, as well as between the last and perchloric or triflic acids.

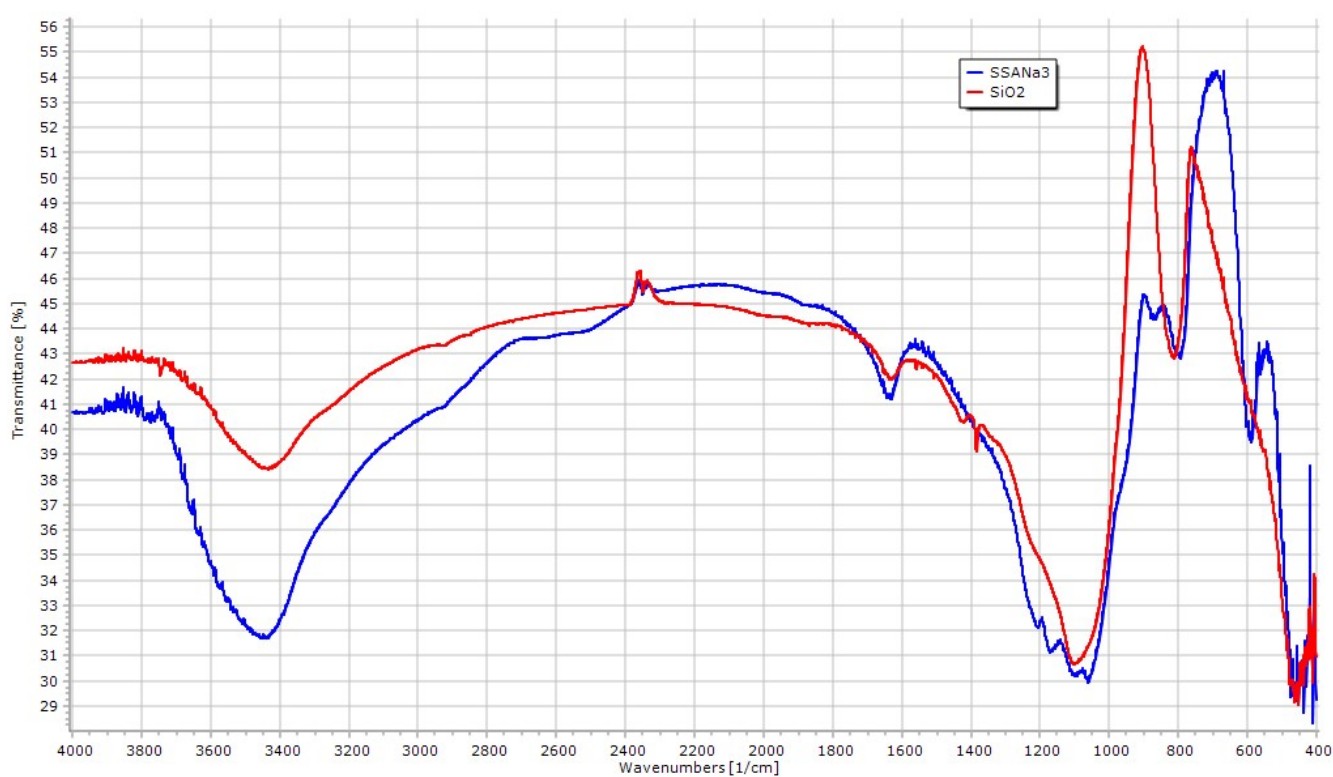

**Figure 4.** IR spectra of SSANa (3.0 mmol/g, blue) vs. SiO$_2$ (red).

**Figure 5.** Proposed model describing the adsorption of sodium bisulfate on silica.

In consideration of the bifunctional acid/base nature of the bisulfate a plausible reaction mechanism is proposed in Figure 6. It involves a series of reversible protona­tion/deprotonation steps, starting from the formation of a non-cyclic hemiacetal between acetone and a primary hydroxyl group and followed by an intramolecular S$_N$1 step [10], that is expected to be faster than other substitution modes [13,42].

In order to study the SSANa-catalyzed process on scale, we became worried about the use of MS inside the reaction mixture, due to the difficulties on the catalyst—desiccant separation stage and due to the observed interaction between **gly** and MS. A solution was to continuously reflux the solvent on the desiccant, kept apart in a modified Dean-Stark apparatus. In analogy of what was obtained with a special gear [49], the inclusion of a long arm funnel inside a standard glassware did the job, (see Figure S1) forcing the solvent to get into contact with the desiccant.

**Figure 6.** Plausible reaction mechanism.

Using a double volume of acetone, to account for the drying function, [16] a respectable 79% yield was obtained after 4 h, but only when a large mass of MS (double, in respect to **gly**) was employed. Moreover, the temperature required to keep acetone at reflux (~65 °C) gave an undesired minor yellowing of the crude product. We then thought that a low boiling solvent able to form a positive azeotrope with water was a convenient way to promptly remove the moisture, if the process could tolerate the consequent dilution. Fortunately, this was the case and adding an adequate volume of co-solvent made the process to work at its best. For example, 25 g of **gly** was converted into **1** in high yield when $CH_2Cl_2$ was added as a co-solvent (150 mL). Its azeotrope with water (boiling at 38.8 °C, 0.4% $H_2O$) let the use of a reduced amount of MS (20 g of 4 Å, 3 mm beads), avoided co-distillation of acetone, and helped the timely completion of the process (4 h, overall). Having an independent (azeotropic) anhydrification technique represents a good advantage, in order to accommodate an increased contamination by water or methanol, typical for glycerol coming from biodiesel plants [15,16], through an adaptation of co-solvent volume/desiccant mass/process time. This system proved to be suitable for scale-up and 100 g of **gly** were converted into solketal with a 96% isolated yield using standard lab glassware (see Section 2.5). In order to further shape the process for industrial operation, future developments could focus on the opportunity to avoid MS and proceed with a water removal uniquely based on the non-miscibility/density difference with co-solvent; even evaluating alternative co-solvents.

Some efforts to recycle the catalyst by simple filtration, drying at 130 °C, and reuse in following reactions failed due to adsorption of organic material on silica, signaled by the browning of the powder during drying. Thermal regeneration at 600 °C left a white powder containing $Na_2SO_4$, coming from pyrosulfate decomposition, with the loss of one equivalent of $SO_3$ (see Figure S4). This allowed the recycle of the catalyst, that was obtained in a fully functional form by the addition of equimolar amount of aqueous $H_2SO_4$ and subsequent overnight drying. This regeneration method was tested in a set of five successive repetitions (100 g scale), showing a sustained high catalytic activity (see Figure S5). Despite the little catalyst cost and its easy preparation make the described regeneration hardly justified in small-scale productions, a milder thermal treatment should be matter of future investigation (e.g., heating at 400–450 °C) in order to retain the starting sulfate as pyrosulfate, with better sustainability. This provides to SSaNa further potential over SSA, considering that $H_2SO_4$ starts its decomposition at 300 °C.

## 4. Conclusions

Working in the large volume, low value sector, **gly** transformation must rely on simple and robust chemistry. In this work a simple to prepare, cheap, and effective catalyst (i.e.,

plain bisulfate on silica, SSANa 3.0 mmol/g) was applied to the conversion of glycerol into its acetone acetal 1, in line with a use-inspired basic research. It features very low toxicity and easy separation from the reaction mixture, at the end of the process. With respect to SSA (3.0 mmol/g), it shows better anchoring on silica and significantly lower acidity, promising less plant corrosion issues in industrial setting and better recyclability. With the assistance of a proper anhydrification technique, this method is able to rapidly convert 100 g of **gly** into solketal, obtaining a high isolated yield (96%) and employing mild process conditions.

**Supplementary Materials:** The following are available online at https://www.mdpi.com/2227-9717/9/1/141/s1, Figure S1: Modified glassware for continuous solvent drying, Figure S2: Crude solketal $^1$H NMR spectra (100 g scale), Figure S3: Distilled solketal $^1$H NMR spectra (100 g scale), Figure S4: Catalyst regeneration cycle, Figure S5: Performance of regenerated catalyst.

**Author Contributions:** Research design, supervision and writing, F.R. Implementation of the research, manuscript correction and writing, L.F. Experimental work and data collection, S.D. Analysis of reaction mixtures and manuscript correction, L.M. All authors have read and agreed to the published version of the manuscript.

**Funding:** This research received no external funding.

**Institutional Review Board Statement:** Not applicable.

**Informed Consent Statement:** Not applicable.

**Data Availability Statement:** Data is contained within the article or supplementary material.

**Conflicts of Interest:** The authors declare no conflict of interest.

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
