# Peer review of "An Expedient Catalytic Process to Obtain Solketal from Biobased Glycerol"

_processes, doi:10.3390/pr9010141_

Round 1
Reviewer 1 Report
In the presented paper authors did a really good job by developing an effective heterogeneous catalyst for glycerol transformation into solketal. This process is of high industrial interest and the results obtained by authors are interesting for the further development of the field.
However, the construction of the article is quite confusing. Thus, the introduction section is far too long and significantly overfilled with general statements. The “results and discussion section” incorporates far too detailed review on the catalytic activity of other catalysts. One would expect to see this description of other catalysts with their general advantages and disadvantages in the introduction, while the discussion part should be devoted to the presentation of authors own results and their comparison with previously described catalysts.
The results of given paper are also not completely clear. It seems that effectiveness of new SSANa catalyst is the same as of simple SSA. Keeping in mind difficulties of SSANa reactivation, the conclusions on the usefulness of this catalysts are questionable.
Based of abovementioned I recommend major revision and greatly encourage authors to reconsider the structure of the article, so it will have clear introduction demonstrating current success in the field, discussion on the properties of SSANa, clear presentation of the advantages and disadvantages of this catalyst in comparison with existing ones (specifically SSA), and conclusions of potential suitability of SSANa for the industrial synthesis of solketal
Reviewer 2 Report
The authors described the catalytic activity of plain bisulfate on silica in the transformation of glycerol into solketal which reached 96%.
In my opinion, the obtained results are interesting but there are a few major and minor issues addressed to the authors of this work.
The main issue is addressed to the conclusion that the best catalyst tested for the studied process is SSANa 3.0mmol/g. But Table 2 shows that all catalysts SPCA, STA, SSAK SSAN which are already reported in the literature give the same results.
Moreover, in the abstract and conclusions, the yield reaches 96% but there is no such value presented in manuscript tables.
Is this high yield was observed only for the 100g scale?
Please explain to me what the improvement was presented in the manuscript method/catalyst compared to the reported systems?
Minor issues are the following:
- Please add to the abstract information of the amount of sulfate supported on 1g gram of SiO2.
- For the glycerol use gly symbol please use bold, italics, or capital letters – manuscript and supporting.
- Line 42 – change vs for vs.
- Figure 1. Shift “R=Me or Et below “ROH”.
- Add to the Introduction the aim of your work. Explain in the text why did you decide to study this topic?
- Line 70 – “2.1 General.” Change for 2.1 General Information.
- In 2.1 General Information assign a supplier to each reagent.
- In 2.1 General Information – info about molecular sieves is missing – supplier.
- In preparation sections please add mmol amounts in all possible places e.g. line 80, 83, 93, 95, 97, 104, etc.
- Line 82 – add 3.0 mmol of what?/g.
- Line 87 – there is no supplier for“MS” described. Description of MS is provided on page 4 but MS is already used on page 3.
- Line 104- bold “1”.
- Table 1 – title – Preliminary evaluation… as of what kind of process?
- Table 1 – change “n.” to Entry.
- Change “a.” to Reaction conditions and then a and b as additional information.
- Change “n. …” to Entry … in the text.
- Figure 1 and 2 – change gly to bold, italic, or capital letters.
- Line 181 – “3.0 mmolacid/g” is confusing. Change to e.g. 3.0 mmol of acid per g of silica.
- Line 182 – kg instead of Kg.
- Figure 3. Add the name of the process. Performance in what process?
- Table 2 – title - Evaluation of different acids in what process?
- change to reaction conditions. Remove b-g symbols. It is obvious that explanations of abbreviations are connected with these in the table.
- Line 226 – vs change to vs.
- Table 1, 2, 3 4 – add an isolated yield of 1.
- Table 3 – please explain “risp.”.
- What do you mean by “anhydrificant”? Is this an English word?
- Table 2 – the letters are cut.
- Table 3 and 4 – remove a. -> Reaction condition.
- Line 169 – “in the acetalization process of Fig. 2.” change to in the acetalization process (Fig. 2).
29 Lines e.g. 169 and 178 -use only one of these two descriptions Fig. X or Figure X.
- Figure 6 – please remove the reaction mechanism since it is not supported by any proof.
- Use in the abstract symbol SSANa 3.0mmol/g beside the description of catalyst.
- Use in Conclusion – plain bisulfate on silica SSANa 3.0mmol/g.
Reviewer 3 Report
There is a need to develop procedures that converts glycerol into bulk products for the industry, such as fuel additives, or intermediates for commodities materials (polymers, paints, etc.) and pharmaceuticals. The finding that NaHSO4 absorbed on silicagel realizes a good catalyst for the formation of solketal (dioxolane from glycerol and acetone) is interesting. The laboratory experiments reported represent a preliminary studies for such a process development. One learns that with an excess of acetone glycerol is acetalized in a good yield in the presence of the new (?) catalyst that is not destroyed by water. At this stage the process requires a large amount of CH2Cl2 for insuring a good yield because water, the co-product of the reaction, must be eliminated by azeotropic distillation. A classical concept. What is not convincing at this stage is that one needs to dry the distilled azeotropic mixture with molecular sieves, an operation that consumes energy (distillation, recovery of the molecular sieves) and which increases the cost of production compared with a simpler elimination of water by the formation of two liquid phases. Why not having assayed other organic solvents such pentane, hexane, cyclohexane, toluene, at normal pressure or reduced pressure? These options may fail because of the co-distillation of acetone (ternary mixtures).
If solketal should be marketed as additive to fuels, how much chlorine does it contains? Traces of chlorine will be bad to engines. Why not use tetrahydrofuran instead of CH2Cl2 for the azeotropic distillation?
The work should be completed answering the questions raised here-above.
One needs to know of the final content (traces) of chlorine in the final product, the yield of recycling of the excess of acetone and the large amount of CH2Cl2. With nowadays regulations it is difficult to see how such a process using CH2Cl2 will be permitted by the authorities of the European Community.
How much CH2Cl2 is not recovered after a number of process cycles?
The propose mechanism of Figure 6 is not plausible. It implies an SN2 displacement of a secondary alcohol moieties by the bulky Me2C(OR)OH group. More reasonable would be to invoke an SN1 mechanism in which the secondary alcohol moiety at C(2) quenches a Me2C(+)-OR intermediate, a highly stabilized oxy substituted tertiary carbenium ion.
line 153: Louis Pasteur
Round 2
Reviewer 1 Report
The paper presented has improved significantly, all major issues are now solved. That is why I suggest publication after some minor issues are addressed:
- Since authors prepared and tested all catalysts from table 2 it is crucial to add the synthesis and test experiments procedures for these catalysts to the materials and methods section
- Page 7 line 242 change “dot” symbol to “middle dot” at SnCl·2H2O
- The resolution of figure 4 is unacceptably poor. Perhaps it is an issue of MDPI pdf builder, nevertheless I suggest checking whether the quality of original figure is good enough. The usage of png. format should help.
Reviewer 2 Report
The authors introduced the proposed changes. Their arguments regarding my doubts are adequate. Therefore, in my opinion, the manuscript can be accepted for publication in the present version.
Author Response
We thank Reviewer 2 for the valuable discussion
kind regards
Reviewer 3 Report
The authors have given very evasive answers to the points I have raised. In chemistry one makes experiments to prove or support what one pretends.
The proton NMR spectra given are cut above 5 ppm, thus one cannot tell whether some CH2Cl2 remains or not in the crude or/and in the distilled final product. There are specific analytical tools to determine trace amounts of Cl in any organic mixture. CH2Cl2 can generate 1,3-dioxolane and 1,3-dioxanes with polyols. Chlorine might be found in the final product under an other form than CH2Cl2. A simple determination of the Cl traces is the best way to make this point clear and convincing. A proton NMR spectrum can hide many impurities. As long as the latter are not chlorinated it is not of serious concern in this context. A proton NMR spectrum with a large signal/noise ratio will make possible to observe the 13C-satellite signals of the two methyl peaks of the acetonide moiety, the area of which can be compared with those of weak signals of some eventual impurities. Sometime this method is more reliable than a chromatographic technique. Elemental analysis can also give a test about product purity.
Concerning the recovery of CH2Cl2 it is clear that a lot of it is lost. One may hope that once the preliminary experiments presented will be translated into an industrial process that this problem will be solved (?).
The authors do not want to use ethereal solvents because they generate peroxides. This is true only if air is introduced in the system. MeOMe and t-BuOMe do not form peroxides. Please note that solketal is also an ether (secondary and tertiary (RO)C-H bonds) and can generate peroxides of it own easily in the presence of air.
